# SHADOWDRAW: FROM ANY OBJECT TO SHADOW–DRAWING COMPOSITIONAL ART

Figure 1: **Generating shadow–drawing compositional art.** Given an arbitrary 3D object, our framework jointly predicts scene parameters, including object pose and lighting, and a partial line drawing, such that the cast shadow seamlessly completes the drawing into a coherent image. The system unites physical shadows with generative drawing, creating compelling compositions from cast shadows that provide only minimal structural cues. Our approach enables straightforward real-world deployment, as demonstrated with physical prototypes of letters I, C, L, R. *Best viewed in Adobe Acrobat Reader for the embedded animation.*

## ABSTRACT

We introduce SHADOWDRAW, a framework that transforms ordinary 3D objects into shadow–drawing compositional art. Given a 3D object, our system predicts scene parameters—including object pose and lighting—together with a partial line drawing, such that the cast shadow completes the drawing into a recognizable image. To this end, we optimize scene configurations to reveal meaningful shadows, employ shadow contours to guide line drawing generation, and adopt automatic evaluation to enforce shadow-drawing coherence and visual quality. Experiments show that SHADOWDRAW produces compelling results across diverse inputs, from real-world scans and curated datasets to generative assets, and naturally extends to multi-object scenes, animations, and physical deployments. Our work provides a practical pipeline for creating shadow–drawing art and broadens the design space of computational visual art, bridging the gap between algorithmic design and artistic storytelling. Check out our project page for more results.

## 1 INTRODUCTION

Shadows have long captivated artists and audiences alike, serving as a powerful medium of expression across cultures and epochs. From traditional Chinese shadow theater, where cut-out puppets project intricate silhouettes, to contemporary shadow photography and installation art that manipulate light to craft evocative narratives (Fig. 2(a)(i–ii)), shadows transform absence into striking imagery. Defined by the shifting interplay of light and form, they are fragile in appearance yet rich in depth, embodying the delicate relationship between illumination, object, and perception.

Recent research in computational visual art has sought to formalize and extend shadow art practices. By framing the problem as an inverse design task, prior works optimize object geometry (Mitra & Pauly, 2009; Sadekar et al., 2022), material properties (Baran et al., 2012; Min et al., 2017), and lighting (Weyrich et al., 2009) to achieve desired visual effects (Fig. 2(a)(iii)). While effective, these approaches typically treat the shadow as an isolated medium, without interaction with other visual elements. Moreover, they assume a predefined target (*i.e.*, knowing a priori what to generate) and rely on parameter optimization to reproduce it.

In this work, in contrast, we are intersted in exploring the delicate interplay between physical shadows and generative drawing. Our inspiration comes from Belgian artist Vincent Bal (Bal, 2025), whose playful works reveal how the cast shadows of everyday objects can seamlessly complete drawn elements. Motivated by this principle, we aim to develop a computational framework that seeks to capture the same sense of serendipity: unified compositions where shadow and line drawing contribute to a cohesive whole, without relying on any predefined target.

Formally, given a 3D object as input, we aim to predict both scene parameters (light and object pose) and a partial line drawing such that, when the scene is illuminated, the shadow cast by the object completes the drawing into a coherent, recognizable image (Fig. 2(b)). Enabling such compositions is however extremely challenging: conditioning a generative model on rendered inputs such as the shadow image or the object–shadow composite provides only weak structural cues, often yielding generated compositions in which the shadow contributes little. This difficulty is further compounded by the scarcity of shadow–drawing examples online, leaving only limited data for training.

To address these challenges, we reformulate the problem around the *shadow contour*—the boundary contour of the shadow. Although a raw shadow and its contour encode the same geometry, reducing the shadow to a clean binary outline yields stronger conditioning: models trained on grayscale shadows often drift from the intended geometry, whereas contour-based conditioning achieves tighter alignment. The shadow contour also naturally supports scalable data construction, as closed contours can be efficiently extracted from generic line drawings, and further enables the use of pretrained edge-conditioned generative models for line drawing generation. Building on this idea, we train a line drawing generator jointly conditioned on the shadow contour and text to ensure consistency with shadow geometry. We then search over scene parameters to reveal distinctive, semantically meaningful shadows, guided by vision–language models that propose line drawing descriptions aligned with the contour. Finally, we curate results using automatic metrics for coherence and visual quality, retaining only those compositions where the shadow meaningfully contributes.

Figure 2: **From traditional shadow art to shadow–drawing compositional art.** (a) Traditional shadow art, such as artist-crafted works and computational designs, treats the shadow as the sole medium. (b) Our framework integrates shadows with line drawings: given a 3D object, we *generate a partial drawing* and *estimate scene parameters* (e.g., object pose and light position) such that the cast shadow completes the composition.

Extensive experiments demonstrate the effectiveness of our framework in generating coherent and visually compelling shadow–drawing compositions. Our approach applies broadly to diverse object models, including real-world scans, curated 3D datasets, and generative assets (Fig. 1), and naturally extends to multi-object scenes, animated settings, and physical deployments (Fig. 5). Notably, the required physical setup is simple: an everyday object, a planar surface, and a movable spotlight suffice to reproduce our computational designs. This accessibility broadens the expressive range of shadow art and lowers the barrier for exploring this form of visual storytelling.

## 2 RELATED WORK

**Computational Visual Art.** Computational visual art explores artistic creations whose perceptual impact depends on 3D geometry, material properties, and controlled illumination. Works span sculpture, architecture, and fabricated objects, often exploiting optical phenomena such as shadow, reflection, or refraction to produce striking imagery (Wu et al., 2022b). Traditionally, the creation process has relied heavily on an artist's intuition and iterative trial-and-error, making it labor-intensive and technically demanding. Recent computational approaches aim to automate or assist this process by formulating it as an inverse problem: given a target visual effect, optimize scene parameters—such as object pose, geometry, and light configuration—so the rendered appearance matches the design intent. Techniques range from combinatorial optimization for occluder placement (Min et al., 2017; Baran et al., 2012) to differentiable rendering for gradient-based shape refinement (Wu et al., 2022a; Sadekar et al., 2022). Applications span reflection-based art (Weyrich et al., 2009), volumetric displays (Hirayama et al., 2019), and multiview illusions (Feng et al., 2024). Unlike these approaches, which optimize scene parameters to achieve a *given* target visual effect, our system *jointly* estimates both the scene parameters and the desired outcome, making the task substantially more challenging.

**Line Drawing Generation.** Line drawings have long been studied as a compact yet expressive representation of shape and semantics. 3D-based approaches derived drawings directly from geometry, extracting contours, depth cues, or neural features to approximate sketches (DeCarlo et al., 2003; Judd et al., 2007; Liu et al., 2020; Choi et al., 2024). In parallel, image-based methods framed line drawing as a supervised translation task, mapping photographs to vector strokes or contours using paired data (Li et al., 2019a;b) or unpaired data (Chan et al., 2022). Beyond image-to-drawing, recent research has explored diverse tasks, including text-guided generation (Frans et al., 2022), object sketching (Liu et al., 2021; Li et al., 2017), portrait rendering (Yi et al., 2019), line drawing completion (Bhunia et al., 2022; Liu et al., 2019), and sequential or collaborative creation (Vinker et al., 2025). Despite these advances, prior studies primarily regard line drawing as an isolated modality, whereas our work explicitly connects it with cast shadows to create hybrid visual compositions.

**Shadow Art.** Shadow art is a subclass of computational visual art where the shadow cast by a physical object under a controlled light source is used for artistic expression. Early computational methods focused on single-light, binary-shadow designs, deforming object geometry so that the resulting shadow matched the desired shape (Mitra & Pauly, 2009; Hsiao et al., 2018). With the advent of differentiable rendering, object shapes can be directly optimized with respect to the shadow image under more sophisticated setups (Sadekar et al., 2022; Qu et al., 2024). However, these approaches may produce irregular or impractical geometries, limiting their real-world applicability. Recent methods have broadened the design space to include multi-layer occluder systems (Min et al., 2017) and color shadow projections using translucent materials (Baran et al., 2012). Beyond rigid objects, shadow art has been explored using human bodies (Won & Lee, 2016) and hand gestures (Xu et al., 2025). In contrast to prior work that treats shadows as the sole visual medium, our approach explores the interplay between such physical effects in 3D space and generative models operating in the pixel domain. Given a candidate cast shadow, which may offer only the barest suggestion of the underlying object, the system must imagine how to complete it into an interesting composition.

Figure 3: **Framework overview.** Given a 3D object, we first optimize scene parameters specifying the object pose and light configuration. From the rendered shadows, we derive text prompts with VLM and extract shadow contours, which together condition the line drawing generator. The generated drawings are then filtered using a VQA-based coherence check and ranked by semantic and quality metrics. The final output is a partial line drawing along with scene parameters that, when rendered, form a coherent shadow–drawing composition.

# 3 GENERATIVE SHADOW-DRAWING ART

We explore an art form that unifies shadow and line drawing into a single, coherent composition. The input is a 3D object model, and the outputs are: (i) a partial line drawing, and (ii) scene parameters specifying the 3D position and direction of a light source and the pose of the object. These elements can be jointly arranged so that, when illuminated, the cast shadow completes the line drawing, producing a recognizable image on the projection surface.

In our setup, the canvas lies on the ground plane, and the light source is modeled as a spotlight to produce sharp, coherent shadows. The light maintains a fixed distance from the canvas center, yielding two degrees of freedom: elevation and azimuth. The object can rotate about its vertical axis and translate along two axes on the ground plane. We normalize the object's longest dimension to standardize shadow size; in physical deployments, the canvas can be inversely scaled to preserve this proportion. The line drawing is rendered in black, and the shadow is represented as a gray silhouette, with the final composition emerging from their precise spatial alignment. For simplicity, we restrict our study to single-light configurations and omit surface textures.

**Overview.** Our task is highly under-constrained: a single 3D object admits countless combinations of light, object, and line drawing arrangements, and traditional practice relies on artistic intuition to achieve compelling results. To make the problem tractable, we first consider a simplified setting where the object pose and light direction are fixed, aiming to generate a line drawing that seamlessly integrates with the shadow. We then extend to optimizing scene parameters to produce visually distinctive and semantically meaningful shadows, accompanied by textual descriptions of the intended subject. Finally, we retain only the best results, selecting those with high visual quality, strong shadow contribution, and tight coherence between the line drawing and shadow.

## 3.1 FROM SHADOW CONTOURS TO SHADOW-DRAWING ART

We begin with a simplified setting where the scene parameters are fixed, the subject description is given, and the goal is to generate a line drawing that integrates seamlessly with the rendered shadow.

A straightforward baseline is to train an image-conditioned generative model on rendered inputs such as the shadow image or the object–shadow composite. However, this approach faces two challenges: weak conditioning signals, which limit the model's ability to align the drawing with the shadow, and data scarcity, as only a few dozen shadow–drawing examples exist online.

To address these issues, we replace the raw shadow with its 2D boundary contour, referred to as the *shadow contour*, as the conditioning input. Although a raw shadow and its contour encode the same geometry, reducing it to a clean binary outline provides stronger conditioning: models trained on grayscale shadows often drift from the intended geometry, whereas contour-based conditioning achieves tighter alignment. Beyond being a strong geometric cue, reformulating the task as transforming closed contours into line drawings yields two benefits: (i) it enables the use of well-established edge-conditioned generative models, and (ii) it allows scalable data synthesis, since shadow-like contours can be efficiently extracted from generic line drawings. As we will demonstrate in the experiments, this shadow contour design significantly improves generation quality.

Our dataset construction pipeline (details and examples in Appendix B.1) proceeds as follows. First, we generate a set of line drawings using GPT-4o and retain only those containing regions bounded by strokes. We then train a FLUX-1-dev LoRA (Labs, 2024) on this filtered set and use it to synthesize an additional 10K line drawings from GPT-4o-generated prompts about everyday subjects. Closed contours extracted from these drawings serve as shadow contour conditions.

Using this dataset, we train a latent flow-based model $\epsilon_\theta$ with the standard score-matching objective:

$$\min_\theta \mathbb{E}_{\mathbf{x}_0, \epsilon, \mathbf{c}_i, \mathbf{c}_t, t} \left\| \omega(t) \left( \epsilon_\theta(\mathbf{x}_t, \mathbf{c}_i, \mathbf{c}_t, t) - \epsilon \right) \right\|^2, \tag{1}$$

where $\mathbf{x}_0$ is the target line drawing latent, $t$ is the timestep, $\mathbf{x}_t$ is the noisy sample at timestep $t$, $\mathbf{c}_i$ is the shadow contour condition, and $\mathbf{c}_t$ is the text prompt. We adopt FLUX.1-Canny-dev (Labs, 2024) as the base model and train a LoRA adapter (Liu et al., 2024) on top.

At inference time, we render the shadow given the scene parameters, extract its boundary, and condition the model jointly on this contour and the text description. To prevent strokes from overlapping the object, we treat generation as an outpainting problem (Lugmayr et al., 2022) with a binary object mask $\mathbf{m}$, preserving masked regions during denoising:

$$\mathbf{x}_t = \mathbf{m} \odot \mathbf{x}_t^{\text{mask}} + (1 - \mathbf{m}) \odot \hat{\mathbf{x}}_t, \quad \mathbf{x}_t^{\text{mask}} \sim \mathcal{N}(\sqrt{\bar{\alpha}_t} x_0^{\text{mask}}, (1 - \bar{\alpha}_t)\mathbf{I}), \tag{2}$$

where $\mathbf{x}_0^{\text{mask}}$ is the latent of the mask $\mathbf{m}$ and $\hat{\mathbf{x}}_t$ is the model prediction at timestep $t$.

Finally, we erase the input shadow contour from the generated drawing, reinsert the 3D object, and render the final composition in which the cast shadow completes the drawing.

## 3.2 Scene Configuration Selection

With the line drawing generation model in place, we next search for scene configurations that produce visually distinctive and semantically meaningful shadows. Each candidate configuration is paired with a text prompt describing the intended subject to condition the generation process.

**Scene Parameter Optimization.** We consider five scene parameters: light azimuth $\theta$, elevation $\phi$, object center position in polar coordinates $(r, \gamma)$, and the object rotation about its vertical axis $\alpha$. To ensure the shadow extends toward the canvas center, we set $\gamma = \theta$ and $r = 0.8\times$ the canvas radius.

We quantify shadow quality using fractal dimension (FD) (Falconer, 2013), which measures contour complexity via multi-granularity box counting; a higher FD indicates more irregular, visually rich shapes. To allow gradient-based optimization, we use a differentiable approximation of FD:

$$\mathcal{L} = -\text{FD}(\mathbf{S}), \quad \mathbf{S} = \text{Renderer}(\theta, \phi, r, \gamma, \alpha), \tag{3}$$

where $\mathbf{S}$ denotes the rendered shadow, obtained via PyTorch3D differentiable silhouette rendering.

Empirically, we initialize the search with $48$ configurations, spanning $12$ azimuths in $30°$ increments and $4$ elevations to vary shadow length, each combined with a random vertical axis rotation to diversify shapes. For each initialization, parameter updates are restricted to its local neighborhood to prevent overlap in the optimization spaces of different starting points.

**Visual Prompt Proposal.** Detailed prompts are known to substantially enhance text-to-image generation quality (Hao et al., 2023), and generic descriptions such as "a man" or "a bird" are often too vague to yield coherent shadow–drawing compositions in our settings. Unlike prior computational art methods that limited to a small set of hand-crafted prompts (Feng et al., 2024; Geng et al., 2024), our objective is to support arbitrary inputs while adapting to diverse shadow geometries. This calls for an automated pipeline that generates scene-specific prompts directly from the shadow itself.

To achieve this, we employ vision-language models. The model is instructed to imagine a line drawing in which the given shadow contour naturally functions as a key structural element, and generate a detailed description of that drawing. To promote reasoning about the stroke's shape and to maintain both semantic and visual coherence, we adopt a chain-of-thought–style prompting template, with the complete system prompt provided in Appendix B.2.

## 3.3 Evaluation and Ranking

From the pool of generated compositions, we apply a systematic filtering process. Each candidate is assessed along three dimensions: (i) *visual quality*, (ii) *shadow–drawing coherence*, and (iii) the

*shadow's contribution* to the overall composition. These criteria ensure that the selected outputs are both visually compelling and structurally coherent. Visual illustrations are shown in Appendix B.4.

**Shadow-drawing Coherence Verification.** We adopt a VQA-based verification strategy (Lin et al., 2024) to measure the coherence between the given stroke and the line drawing. During the Visual Prompt Proposal stage, the VLM is instructed to specify the intended role of the shadow contour in the composition (e.g., "the body of a fish"). We then overlay the shadow contour in red onto the generated line drawing and query another VLM with a yes/no question: *"Does the highlighted stroke outline the described component?"* Candidates receiving a "no" response are discarded.

**Shadow Contribution Assessment.** We further evaluate whether the shadow contributes positively to the composition. Specifically, we compare the complete line drawing (*full*) with a version where the shadow contour is removed (*partial*), using CLIP text–image similarity (Radford et al., 2021), ImageReward (Xu et al., 2023), and Human Preference Score (HPS) (Wu et al., 2023). If the partial version achieves a higher ImageReward or HPS score, the composition is rejected, indicating that the shadow does not enhance the final result.

**Ranking.** For the remaining candidates, we compute an improvement score for each metric:

$$\Delta_{\text{CLIP}} = \frac{\text{CLIP}_{\text{full}}^2}{\text{CLIP}_{\text{partial}}^2}, \; \Delta_{\text{IR}} = \Phi(\text{IR}_{\text{full}})^2 - \Phi(\text{IR}_{\text{partial}})^2, \; \Delta_{\text{HPS}} = \text{HPS}_{\text{full}}^2 - \text{HPS}_{\text{partial}}^2, \quad (4)$$

where $\Phi(\cdot)$ denotes the CDF of the standard Gaussian, as ImageReward scores are normalized. The overall ranking score is defined as:

$$\mathcal{R} = \Delta_{\text{CLIP}} \cdot \Delta_{\text{IR}} \cdot \Delta_{\text{HPS}}, \quad (5)$$

and the top-$K$ compositions by $\mathcal{R}$ are selected as final outputs.

## 4 EXPERIMENTS

In this section, we first detail our experimental setup, then present quantitative comparisons with baselines and ablation variants, followed by qualitative results across diverse 3D assets. Finally, we demonstrate downstream applications that our framework enables out of the box.

### 4.1 EXPERIMENTAL SETUP

**Baselines.** Since no existing method is explicitly designed for generating shadow–drawing compositional art, we construct two baselines using state-of-the-art image generation models. The first, *Gemini (object–shadow)*, employs Gemini Flash 2.5 Image Preview (Google, 2025) to generate the composition conditioned on both the object–shadow composite image and the text prompt produced by our approach. The second, *Gemini (shadow contour)*, replaces the object-shadow composite with the shadow contour image, providing more precise geometric guidance. Details of baseline execution are provided in Appendix B.3. To analyze the role of individual framework components, we conduct ablation studies: (i) training the line drawing generation model on artist-sourced images conditioned on the object–shadow composite, (ii) training the same model conditioned on shadow contour, and (iii) initializing scene parameters randomly without optimization.

**Data.** We collect 200 object models from diverse sources to evaluate our framework (examples shown in Fig. 4): 26 alphabet models (A–Z), 20 from the YCB robotics dataset (Calli et al., 2017), 20 real-world household objects scanned with Polycam (a mobile app), 17 synthetic assets generated by MeshLRM (Wei et al., 2024), 87 objects from Objaverse-LVIS of distinct categories (Deitke et al., 2023), and 30 characters from Objaverse. For training the ablation baselines without our synthetic data, we collect 71 images from the artist's YouTube channel (Bal, 2025), then manually filtered and post-processed in Photoshop to extract the object, shadow, and line drawing components.

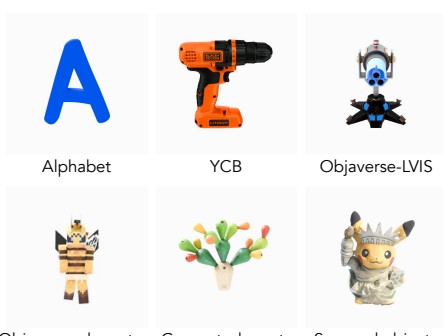

Figure 4: Examples of objects for evaluation.

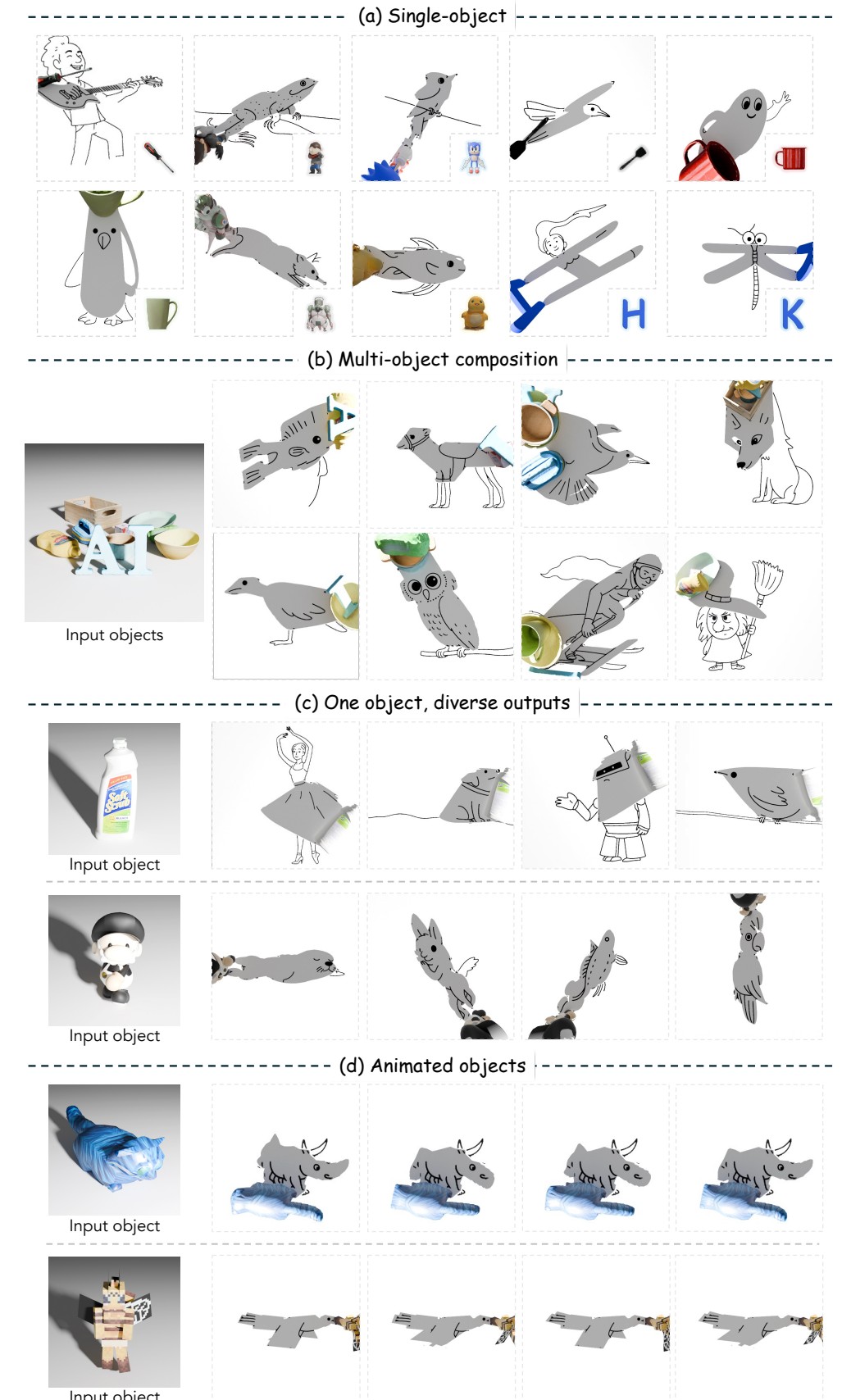

Figure 5: **Gallery of shadow-drawing art generation results.** (a) Single-object generation. (b) Multi-object compositions. (c) Diverse results from the same object by varying light, pose, and line drawing. (d) Animated shadow–drawing art, where the shadow evolves with the motion of the object to complete the composition. More results are available on our project page.

| Method | Condition type | Training data | Scene param optimization | CLIP↑ | Conceal↑ | IR↑ | HPS↑ |
|--------|----------------|---------------|--------------------------|-------|----------|-----|------|
| Ablation 1 | object-shadow | artist source | ✓ | 31.04 | 0.225 | -0.0720 | 0.2244 |
| Ablation 2 | shadow contour | artist source | ✓ | 31.38 | 2.215 | 0.1552 | 0.2269 |
| Ablation 3 | shadow contour | synthetic | ✗ | 32.08 | 2.606 | 0.4177 | 0.2294 |
| **Ours** | shadow contour | synthetic | ✓ | **32.41** | **3.006** | **0.4441** | **0.2373** |

Table 2: **Quantitative ablation studies.** The proposed shadow contour conditioning, synthetic training data, and scene parameter optimization each contribute significantly to the overall generation quality. IR stands for the ImageReward score (Xu et al., 2023), while HPS refers to Human Preference Score (Wu et al., 2023).

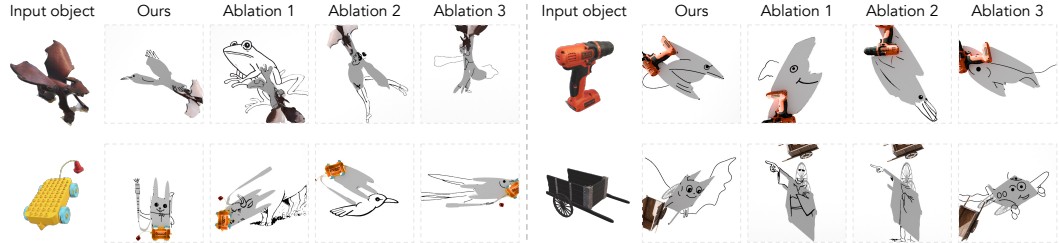

Figure 6: **Qualitative ablation studies.** While ablated variants often yield results where the shadow contributes little, our method produces compositions with better visual quality and stronger shadow–drawing coherence.

**Metrics.** For quantitative evaluation, we compare our method against the baselines among the top-4 ranked outputs. Specifically, we report the average CLIP score (Hessel et al., 2021), the ImageReward score (Xu et al., 2023), and the Human Preference Score (HPS) (Wu et al., 2023) of the generated shadow–drawing compositions. We further measure concealment (Geng et al., 2024), defined as the difference between the CLIP score of the complete line drawing and that of the version with the shadow contour removed. In addition, we conduct a user study to assess human preference: 10 participants are shown the top-1 ranked output of our method and those from the baselines, and asked to select the better result or indicate that neither is preferable.

## 4.2 RESULTS AND ANALYSES

**Baseline Comparisons.** Tab. 1 summarizes comparisons between our framework and the baselines. In the Gemini (object-shadow) setting, the composite image provides only weak structural cues, which are insufficient for producing coherent shadow–drawing compositions. The shadow contour representation offers stronger geometric cues, yet still falls

| Method | CLIP↑ | Conceal↑ | User prefer. ratio (%) |
|--------|-------|----------|------------------------|
| Gemini (object-shadow) | 31.28 | -0.2840 | 3.6% |
| Gemini (shadow contour) | 31.65 | 0.2421 | 6.0% |
| Ours | **32.41** | **3.0059** | **70.4%** |

Table 1: Comparison with the baselines.

short: pretrained image generators struggle to capture the subtle interplay of shadows and drawings. Consequently, both baselines yield low concealment scores, indicating that the shadow contributes little—or even negatively—to the final composition. Consistently, user studies show that participants overwhelmingly favored our method (70.4%), with 20.1% of cases marked as "neither preferable." In contrast, our approach achieves substantially higher concealment scores, confirming that the shadow functions as a crucial structural element rather than a redundant component. Qualitative comparisons in Fig. 7 further highlight the superior visual quality of our approach.

**Ablation Studies.** We conduct ablation studies to evaluate the contributions of three key components: the proposed shadow contour condition, the synthetic training dataset, and scene parameter optimization. As shown in Tab. 2, each component yields clear gains. Replacing object–shadow conditioning with shadow contour notably improves both generation quality and concealment; substituting the limited artist-sourced data with our large-scale synthetic dataset further boosts all metrics; and incorporating scene parameter optimization delivers the strongest overall performance. Qualitative comparisons in Fig. 6 illustrate these effects. A user study further shows that 96.8% of the top-4 images selected by our evaluation and ranking pipeline contain at least one rated satisfactory, validating the framework's effectiveness in producing coherent shadow–drawing compositions. Additional discussions of the evaluation algorithm are provided in Appendix B.4.

Figure 7: **Qualitative baseline comparisons.** Large models like Gemini, though powerful, fail to capture the subtle notion of shadow–drawing art and often produce outputs where the shadow contributes little, whereas our method yields coherent shadow–drawing compositions of better quality.

## 4.3 APPLICATIONS

We demonstrate the versatility of our framework through four applications: (i) generating diverse shadow-drawing art from a single object, (ii) creating shadow-drawing art from multiple input objects, (iii) extending to animated objects, and (iv) deploying in real-world setups.

**One Object, Diverse Results.** Our framework by design generates multiple shadow-drawing compositions from the same real-world object. By varying the light direction, the object pose, and the underlying line drawing, we obtain a collection of artworks that highlight different aspects of the same object. This demonstrates how a single object can serve as the basis for a wide range of artistic expressions. Qualitative examples are shown in Fig. 5(c).

**Multi-object Compositions.** Our framework naturally extends to scenes involving multiple objects. For each candidate configuration, we independently sample self-rotation angles, arrange the objects vertically, and release them in Blender's physics simulation to obtain a stable stacked layout. Once equilibrium is reached, the configuration is treated as a single composite object, allowing the rest of the pipeline to be applied directly. This enables more elaborate results where different objects contribute complementary shadow structures. See Fig. 5(b) for examples.

**Animated Shadow-drawing Art.** Our framework also supports animated objects without extra training. For each configuration, we render five key frames and overlay their shadow contours into a single image, using distinct colors to denote frames. This composite is then fed to the VLM to generate the corresponding prompt. As in the static-object setting, we apply a binary mask to restrict stroke placement, defined as the intersection of all shadow regions and their neighborhoods, to avoid strokes in dynamically changing areas. Details are provided in Appendix B.5. We evaluate this pipeline on animated objects from Objaverse, with qualitative results in Fig. 5(d) and Fig. 10, demonstrating the ability of our system to handle temporally varying scenes.

**Real-world Deployment.** Our method can be readily reproduced in physical settings without specialized equipment, requiring only a 3D object and a single spotlight. In practice, everyday household items combined with a phone flashlight are sufficient to create compelling shadow–drawing compositions. This accessibility positions our framework as a practical tool for artists, educators, and hobbyists, lowering the barrier to exploring computational shadow art. Fig. 1 shows physical prototypes of the letters I, C, L, R, with a complete demonstration available on the project page.

## 5 CONCLUSION

We introduce SHADOWDRAW, a framework for creating unified shadow–drawing compositional art from arbitrary 3D objects. Our approach optimizes scene parameters to discover semantically meaningful configurations, employs shadow contours to guide line drawing generation, and incorporates automatic evaluation and ranking to ensure shadow-drawing coherence and visual quality. Experiments show strong results across diverse 3D assets, with natural extensions to multi-object scenes, animations, and real-world setups. By broadening the design space of computational visual art, our work opens new avenues for accessible, democratized creation of shadow-based art.

**Limitations.** Although our framework consistently produces compelling results across diverse objects, challenges remain. Some objects naturally cast shadows that are visually ambiguous or insufficiently informative, making high-quality compositions difficult. The search over scene parameters also increases runtime, and while our automated ranking surfaces strong candidates, occasional human judgment is still needed to identify the best result. We elaborate these points in Appendix B.7.

## USE OF LARGE LANGUAGE MODELS

We used large language models (LLMs) solely as a writing assist tool to polish the grammar and clarity of the manuscript. LLMs were not involved in research ideation, methodological design, or result analysis.

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

## A    SUPPLEMENTARY MATERIAL OVERVIEW

In this supplementary material, we provide additional implementation details and present extended qualitative results. An anonymous project page is available at project page; a copy is also included in the supplementary file for offline viewing without an Internet connection.

## B    IMPLEMENTATION DETAILS

### B.1    TRAINING DATA CONSTRUCTION

We construct a paired dataset of shadow contours and line drawings to train our line drawing generation model. The pipeline proceeds as follows. First, we generate 100 line drawings using GPT-4o and retain only those that contain closed regions bounded by strokes, using prompts that describe everyday objects. We then train a FLUX-1-dev LoRA (Labs, 2024) on this filtered set and use it to synthesize an additional 10K line drawings from GPT-4o-generated prompts about everyday subjects. Finally, we apply the `FindContour` algorithm in OpenCV to extract closed regions from the drawings, and use a greedy merging algorithm to iteratively combine them until only four regions remain. The closed contours of these regions (and there combination) serve as the shadow contour conditions.

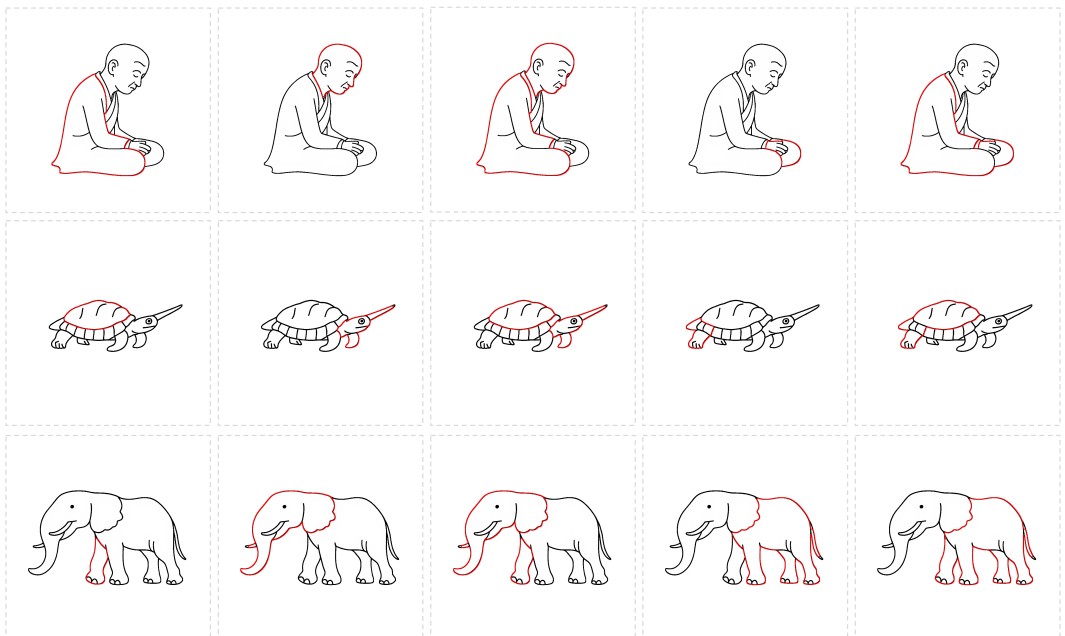

Figure 8: Examples of training data pairs. Each row shows a line drawing generated by our finetuned FLUX model, with different closed contours extracted from it. Each image forms a training pair, where the red contour is used as the condition and the full line drawing serves as the target.

### B.2    VISUAL PROMPT PROPOSAL

Vision–language models are highly sensitive to input formatting, and poorly structured inputs often result in uninformative or inconsistent outputs. To address this, we carefully design a system prompt that guides the VLM, specifically GPT-4.1, to generate detailed, semantically meaningful descriptions of the provided stroke and its role in the complete line drawing. The full system prompt is given below.

```
You are a skilled artist specializing in expressive, imaginative, and
    visually striking minimalist line drawings. You will be shown an
    image containing a contour. Your task is to interpret this contour
    and create a complete line drawing, using the provided contour as the
```

```
   core expressive element of your composition. The subject you draw
   should be a character, either a human, an animal, or a cartoon or
   anthropomorphic figure.

# Instructions

First, analyze the contour to identify its function in the drawing.
   Follow these steps:

1. Analyze the contour's geometry and position on the canvas.

2. Determine the subject of the line drawing and which major, prominent
   body part (such as body, head, face) or clothing (such as skirt or
   dress) the contour outlines. Do not describe small or less essential
   features, such as hands, tails, wings, or beaks. Specify the subject
   in precise terms:
For people, use identifiers (e.g., man, woman) or vocations (e.g., dancer
   , sailor, guitarist);
For animals, name the species (e.g., bird, fish, cat, dog);
For cartoon or anthropomorphic characters, name the type (e.g., ghost,
   robot, cookie character, book character).

3. Explain your reasoning in detail, including the stroke's shape, its
   position on the canvas, and why it is a good fit for the composition.

Next, write a description of the complete drawing without referencing the
    provided contour, following this structure:

1. Opening: A minimalist line drawing of a [character] [in a pose or with
    an expression], matching your earlier interpretation.

2. Physical description: The [character] has [facial feature or
   expression] and wears [clothing or accessories].

3. Object or motion (optional): The [character] is [doing something,
   holding something, or in motion].

4. Gesture or interaction (optional): Further describe the subject's
   gesture or interaction with their environment.

5. Conclude with a style remark: The style is [adjective(s)], [additional
    notes about technique or focus].

# Format requirement

1. Separate the two parts with a blank line.

2. Do not use numbering, bullet points, or extra formatting.

3. Strictly follow this structure without additional comments:

The provided contour shows an outline of the [specific body part or
   clothing] of a [character]. The reason is [contour geometry
   interpretation]. [Additional reasons for your interpretation].

A minimalist line drawing of a [character] [in a pose or with an
   expression]. The [character] has [facial feature or expression] and
   wears [clothing/accessories]. [Optional action or motion]. [Optional
   gesture or interaction]. [Artistic style remark].
```

Listing 1: System prompt for creating the textual description of the intended line drawing in the visual prompt proposal stage.

### B.3 BASELINE EXECUTION DETAILS

Here we describe how we use Gemini Flash 2.5 Image Preview (a.k.a. nano banana) (Google, 2025) to generate shadow–drawing art. In the *object–shadow* version, we provide the model with the object–shadow composite and the line drawing description produced by our approach, and ask it to directly generate a shadow–drawing composition. In the *shadow contour* version, we instead provide the shadow contour and the line drawing description, prompting the model to complete the drawing. As in our framework, we then remove the input shadow contour from the generated drawing, reinsert the 3D object, and render the final composition, where the cast shadow completes the drawing.

### B.4 DISCUSSION ON THE EVALUATION ALGORITHM

**Visual Illustration.** Figure 9 illustrates how our evaluation and ranking algorithm selects high-quality shadow-drawing compositions. In the first stage, the VQA-based verification discards incoherent cases (e.g., when the shadow stroke does not correspond to the intended body part of the character). In the second stage, the shadow contribution assessment compares complete and contour-removed versions, ensuring that the shadow meaningfully enhances the drawing. As shown, this two-step process ranks plausible results higher while filtering out cases where the shadow plays only a minor or misleading role. Overall, the pipeline balances semantic alignment, structural coherence, and visual quality, producing consistent and interpretable rankings.

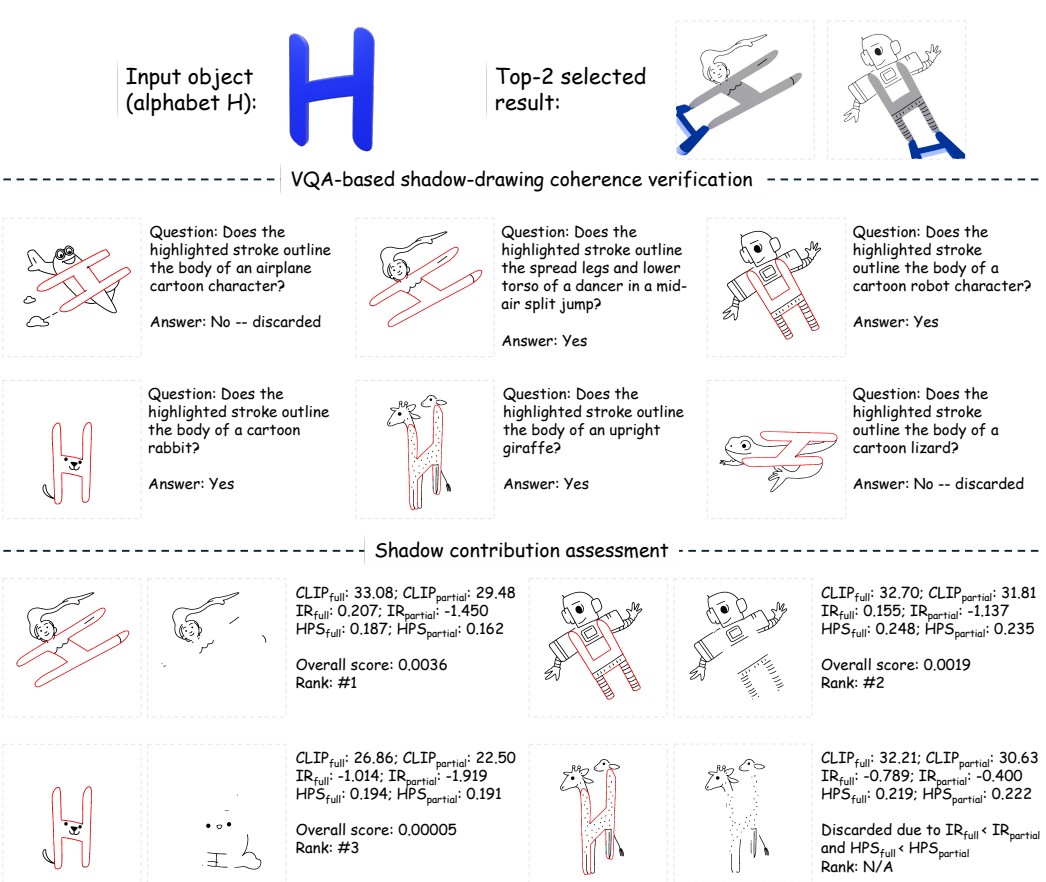

Figure 9: Illustration of the evaluation and ranking process, which discards incoherent cases and preserves only those where the shadow meaningfully contributes to the composition.

**Analyses.** Evaluating our generated shadow–drawing compositions is inherently challenging, as their abstract and artistic qualities often resist objective evaluation. To rigorously assess effectiveness, we designed two complementary user studies based on pairwise preference judgments. (1)

For each object, we randomly select one result from the top-4 ranked outputs and another from the remaining, asking evaluators to choose their preferred composition; and (2) we randomly select two results among the top-3, where differences are more subtle. In both cases, evaluators may also indicate that neither is preferable. Altogether, we collected 2,000 preference pairs from 10 annotators for the first study, and 2 labels per top-3 pair across all 200 objects for the second study.

In the first study, our ranking algorithm achieves a strong alignment with human judgment, agreeing on 63.5% of pairs, disagreeing on only 11.0%, and with 25.5% of cases marked as no clear preference. These results indicate that our automated ranking provides a reliable proxy for subjective evaluation in the broader design space. In the second study, where all candidates are already of very high quality, the agreement rate with human preference naturally decreases to 39.8%, with another 24.3% of cases judged as indeterminate. Crucially, the agreement between two independent sets of human annotations is itself only 44.5%, underscoring the intrinsic subjectivity of evaluating artistic compositions. Taken together, these findings suggest that while perfect alignment is unattainable in such a subjective domain, our ranking system performs comparably to human consensus and thus provides a practical, scalable tool for curating shadow–drawing art.

### B.5 Animated Shadow-drawing Art

As mentioned in Sec. 4.3, our framework supports animated objects without requiring additional training. We provide the implementation details as follows. For each candidate configuration, we render five keyframes of the animation and extract their shadow contours. To preserve temporal information, we overlay the strokes into a single composite image, assigning distinct colors to each frame so that the VLM can recognize temporal variation and generate a corresponding prompt.

A critical step in this pipeline is the construction of a binary mask to restrict where strokes may be placed. Without such a constraint, strokes might appear in regions where shadows change across frames, leading to incoherent results. To build the mask, we proceed as follows. First, we render the shadow silhouettes of all five keyframes. Pixels that are covered by shadows in every frame are designated as the *static region*, while pixels that are covered in at least one but not all frames are designated as the *dynamic region*. Next, for each pixel on the canvas, we compute its distance to both the static and dynamic regions. If a pixel is closer to the dynamic region than to the static region, we mask it out, prohibiting stroke placement in that location. Intuitively, this rule ensures that strokes are confined to stable shadow regions and avoid areas subject to temporal variation.

Once the mask is established, the generation process follows the same procedure as in the static-object setting. The shadow contour and the corresponding VLM-generated prompt are fed to the line drawing generator, and masked regions are excluded during synthesis. Finally, the animated object is reinserted into the scene and rendered across frames, with its shadow complementing the static line drawing to form a temporally consistent shadow–drawing composition.

We evaluate this pipeline using animated objects from Objaverse. Qualitative results are presented in Fig. 5(d) and Fig. B.5, with more examples in our project page. As shown, our system successfully generates line drawings that remain coherent with dynamic shadows, demonstrating the ability of our approach to extend from static to temporally varying scenes.

### B.6 Runtime Analysis

The only trainable component in our framework is the line drawing generation model, which we trained for approximately 12 hours on 8 A6000 GPUs. At inference, the dominant cost arises from the diffusion process, taking roughly 40 seconds per image. For a single object instance, running line drawing generation across 48 sampled scene configurations requires about 30 minutes with 30 inference steps, and the entire pipeline completes in under 40 minutes on a single A6000 GPU. Reducing the inference steps from 30 to 10 lowers the latency to around 20 minutes, with little impact on the final result quality.

### B.7 Limitations

While our method enables diverse and visually engaging shadow–drawing compositions, several limitations remain. First, the quality of results is closely tied to the intrinsic shape of the object:

Figure 10: **Animated shadow–drawing art.** Line drawings generated with our pipeline remain coherent as dynamic shadows complete the composition across frames. *Best viewed in Adobe Acrobat Reader for the embedded animation.*

some objects inherently produce shadows that are either visually uninteresting or too ambiguous to interpret, regardless of lighting or pose. As illustrated in Fig. 11, such cases often yield shadows that lack recognizable structure or fail to align meaningfully with the generated drawing. Second, the joint search and generation process over scene parameters introduces noticeable runtime overhead. Although this procedure is necessary to explore the large design space, generating results for a single object currently requires about 40 minutes (see Appendix B.6). Finally, while our ranking algorithm is generally effective at surfacing strong candidates, it is not flawless. In practice, users may still need to examine multiple outputs to identify the most compelling result. Addressing these limitations—through richer shadow descriptors, more efficient search strategies, and refined ranking or user-in-the-loop mechanisms—represents promising directions for future work.

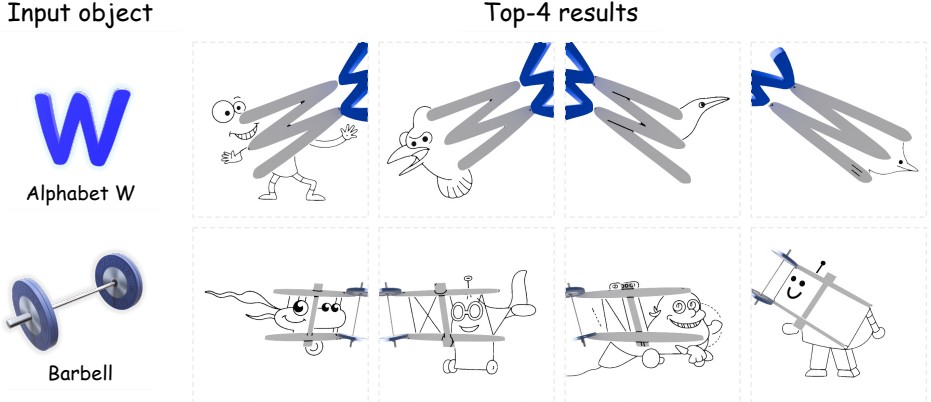

Figure 11: **Failure cases.** Some objects produce shadows that are ambiguous or uninformative, making it difficult for our system to produce meaningful shadow-drawing compositions.

