# OpenReview forum: "ShadowDraw: From Any Object to Shadow–Drawing Compositional Art"
_ICLR.cc/2026/Conference — ICLR 2026 Conference Withdrawn Submission_

### Official Review · Reviewer_Cdhm · 2025-11-01

**Soundness:** 2
**Presentation:** 2
**Contribution:** 2
**Rating:** 4
**Confidence:** 3

**Summary:**

This paper presents a framework for generating compositional art using shadows. Shadows cast by 3D objects are used to complete partial line drawings and represent a concept as a whole. Parameters like lighting, pose are optimized and line drawings are generated conditioned on contours of shadows.

**Strengths:**

- The paper introduces a novel problem statement, finding novel concepts from shadows and drawing them out.
- Using boundary contours rather than raw shadows as conditioning provides stronger geometric guidance

**Weaknesses:**

- The paper lacks significant novelty. The paper chains different networks from data generation to training with limited novelty. While the shadow contouring is effective, as a primary contribution it is not significant. The idea to draw lines on shadows from real objects is quite niche and from a storytelling point of view.
- The paper lacks clarity in many places. Ranking section in 3.3 can be further explained and user studies lack sufficient explanation in the main paper. The contents of the paper are also scattered in appendix and main paper with no proper structure. (eg. user studies referenced in results and analyses but described in appendix)
- The time sink of the entire pipeline from object to generated drawing is 40 mins (important detail, should be mentioned in main paper and not appendix). This is a prohibitively large duration for a relatively simple task. While there might be a weak question on whether this kind of automated shadow drawing is required, the time spent computing does not seem to justify the output.

**Questions:**

- Why did the authors choose Flux-Canny ? It seems a particularly complex network for simple line drawings.
- What do the authors mean by a "greedy merging algorithm"?
- Is there any difference between S and c_i in Eq. 3 and 1 respectively?

---

### Official Review · Reviewer_7MHK · 2025-11-01

**Soundness:** 3
**Presentation:** 3
**Contribution:** 3
**Rating:** 6
**Confidence:** 4

**Summary:**

This paper proposes a method to create line drawings with an effect called "shadow drawing". The shadow is cast by an 3D object and becomes a part of the drawing.

**Strengths:**

1.	This is an interesting problem. It needs some efforts in considering the optical modeling to achieve such effects.
2.	The  application looks satisfying and I will like it when I have a “shadow drawing” in my office.
3.	The technical design looks rational and  those components like line drawing processing and image/lineart completion are explainable and will really achieve the wanted effects.

**Weaknesses:**

1.	I am not sure if this work fits ICLR. From my perspective this paper looks typical for SIGGRAPH/ASIA/TOG. Nevertheless, if other reviewers think it is okay for ICLR scope then we can see this as a kind of “learning representation”.
2.	From the technical standard of ICLR, the main uniqueness of this problem is likely an inpainting task to find where and what to inpaint a line drawing. Although the shape needs to match an 3D object shadow, the pure technical aspect seems very narrow considering the image editing capability of current SOTA image models.
3.	However I can still understand the problem value and some HCI value in the evaluation. But I am not sure the level of contribution in machine learning audiences. I will wait for other reviews and inputs.

**Questions:**

1. What CLIP model is used for CLIP score? There are many better CLIPs like SigLIP2 that can replace the CLIP score for better evaluation.
2. I would see more details about the "Human Preference Score", like the user instructions and user grouping/counting...

---

### Official Review · Reviewer_AGLE · 2025-11-01

**Soundness:** 3
**Presentation:** 4
**Contribution:** 2
**Rating:** 6
**Confidence:** 4

**Summary:**

This paper presents an interesting framework that creates shadow–drawing art from everyday 3D objects. Given any real-world object, the system jointly predicts the object pose, light source configuration, and a partial sketch that completes the composition, such that the cast shadow naturally integrates with the drawing. To enhance the generative model’s ability to produce coherent sketches, the authors propose conditioning on the shadow contour—the binary outline of the shadow—instead of using raw shadow images or object–shadow composites. The methodology is clearly described, combining scene parameter optimization, line drawing generation, and automatic evaluation to ensure visual and semantic coherence. For evaluation, the authors compare their approach with Gemini Flash 2.5 under different conditioning settings, demonstrating a significantly higher human preference score and better visual quality across various objects.

**Strengths:**

- The paper is clearly written and well structured.
- The paper introduces a novel and well-motivated problem—generating shadow–drawing compositional art from arbitrary 3D objects—which extends the frontier of computational visual art and connects physical rendering with generative modeling in an elegant way.
- The proposed framework is technically sound and well engineered, combining shadow contour conditioning, differentiable scene parameter optimization, and automatic evaluation in a coherent pipeline. Each component is justified and shown to contribute meaningfully through ablations.
- The use of shadow contours as conditioning improves geometric alignment and data scalability, addressing a core limitation of prior image- or composite-based conditioning approaches.
- The experimental results show a predominant user preference compared to the baseline methods.

**Weaknesses:**

- While the paper presents an elegant and creative system, the core contributions lie primarily in system design and integration for artistic creation, rather than introducing new scientific, algorithmic, or engineering advances.
- The method relies on having accurate 3D object models (digital twins) in the Scene Configuration Selection stage, which may limit its applicability in casual or unstructured real-world settings where high-quality geometry is unavailable.
- The runtime is relatively long, as generating one final composition requires substantial computation per object, reducing scalability for practical or interactive applications.

**Questions:**

- How practical is the method for real-world scenarios involving everyday objects? Does it require a full and accurate 3D reconstruction of each object before generating the shadow–drawing composition, or could it generalize to partial scans or single-view captures?

- In the scene parameter optimization stage, how is the search space of lighting and object configurations defined and constrained? Specifically, how do the authors ensure that the optimized light positions remain physically plausible and easily reproducible in real-world setups?

---

### Official Review · Reviewer_f574 · 2025-11-06

**Soundness:** 3
**Presentation:** 3
**Contribution:** 2
**Rating:** 4
**Confidence:** 3

**Summary:**

This paper introduces a system that automatically creates shadow-drawing compositions — line drawings whose meaning is completed by the cast shadow of an arbitrary 3D object. The pipeline jointly predicts (i) a stylized line drawing and (ii) scene parameters (object pose and lighting) such that the resulting shadow aligns with and completes the drawing.
The method integrates:
* a shadow-contour–conditioned diffusion model (LoRA on FLUX) trained on synthetic contour→drawing pairs,
* a differentiable scene optimizer maximizing fractal dimension of the shadow contour to encourage visually rich silhouettes,
* a VLM-based prompt generator that proposes textual concepts matching each contour,
* and automated ranking/filtering using CLIP, ImageReward, Human Preference Score, and a concealment metric.
Qualitative results show creative and often convincing “shadow-drawing” compositions, including multi-object and animated scenes, and physical prototypes.

**Strengths:**

* Novel problem formulation. Casting “shadow-drawing art” as a computational generation task is original and visually compelling.
* End-to-end system design. Integrates differentiable rendering, optimization, text-vision prompting, and diffusion-based generation into a cohesive automated pipeline.
* Empirical insight: Demonstrates that closed shadow contours yield stronger conditioning than grayscale shadows or composites for contour-to-drawing generation.
* Synthetic data pipeline. Clever use of LLMs and a LoRA diffusion model to synthesize large contour–drawing pairs when artist data is scarce.
* Differentiable fractal-dimension objective. Creative heuristic to search lighting/pose configurations that produce visually interesting shadows.
* Automated evaluation and ranking. Uses learned reward metrics (ImageReward, HPS) to filter appealing results; this produces consistent quality.
* Solid visuals. Includes single/multi-object, animation, and real-world setups demonstrating generality and practical creativity.

**Weaknesses:**

* I’m unsure if there is sufficient new learning for a venue like ICLR. Core algorithmic components (diffusion, LoRA, VLM prompting, differentiable rendering) are standard; main contribution lies in engineering and system integration rather than a new learning principle or model.

* What is the relationship between the number of sampled views (I believe 48 is default) and success rate? Can users supply a target view/light source and get line drawings that would be a good fit?

* The lack of control is a concern for an artist expression tool like this. The user can pick the 3D model, but it does’t appear they can pick the contour drawing of interest or the text prompt - several text prompts are sampled and an automated VQA scoring pipeline filters candidates. Thus there is limited controllability or guarantee of semantic adherence.

* The reported inference costs are quite slow to be useful in practice, requiring 20-40 mins per input object without guarantee that the synthesized results are acceptable to the user.

* This is primarily a creative-system contribution; I'm unsure if there is a theoretical or generalizable ML insight expected for this venue.

* Analysis of the text prompt: what is the sensitivity of the line art generator to the level of details in the text prompt? The authors could consider evaluating this generative model’s performance independent of the full system (e.g., conditioning accuracy, FID, CLIP alignment, prompt fidelity, or diversity).

* In addition, there is an opportunity for learning-based scene optimization, which could elevate the contribution further. Instead of heuristic sampling, a model could be trained to predict the optimal view/light configuration given a 3D object and (optionally) a target drawing or contour. This could accelerate the inference process and add more user control.

**Questions:**

See Weaknesses

---

### Note · Authors · 2025-11-12

I have read and agree with the venue's withdrawal policy on behalf of myself and my co-authors.